# Effect of Low- and Moderate-Intensity Aerobic Training on Body Composition Cardiorespiratory Functions, Biochemical Risk Factors and Adipokines in Morbid Obesity

**DOI:** 10.3390/nu16234251

**Published:** 2024-12-09

**Authors:** Judit Horváth, Ildikó Seres, György Paragh, Péter Fülöp, Zoltán Jenei

**Affiliations:** 1Department of Physical Medicine and Rehabilitation, Faculty of Medicine, University of Debrecen, 4026 Debrecen, Hungary; jenei.zoltan@med.uideb.hu; 2Division of Metabolic Diseases, Department of Internal Medicine, Faculty of Medicine, University of Debrecen, 4032 Debrecen, Hungary; seres@belklinika.com (I.S.); paragh@belklinika.com (G.P.); pfulop@belklinika.com (P.F.)

**Keywords:** obesity, adipokine, low and moderate intensity, aerobic training, cardiorespiratory function, body composition

## Abstract

Background: Obesity poses an enormous public health and economic burden worldwide. Visceral fat accumulation is associated with various metabolic and cardiovascular consequences, resulting in an increased prevalence of atherosclerotic conditions. We aimed to examine the impact of low-and moderate-intensity aerobic training on several anthropometric and cardiorespiratory parameters and markers of atherosclerosis, including inflammation, serum levels of lipoproteins and adipokines of extremely obese patients in poor condition. Methods: Forty severely obese patients were recruited and randomized into two groups, Group 1 and Group 2, for a six-week inpatient study. Group 1 received moderate-intensity (40–60% heart rate reserve) and Group 2 received low-intensity (30–39% of heart rate reserve) aerobic training combined with resistance training. The patients’ cardiorespiratory functions were assessed by ergospirometry. Anthropometric data were recorded, body composition was analyzed and functional tests were performed. We also investigated serum lipids and high-sensitive C-reactive protein levels and calculated the homeostatic model assessment-insulin resistance indices and adipokine levels as predictive biomarkers. Results: Functional abilities and some biochemical parameters, such as homeostatic model assessment-insulin resistance, serum lipids, apolipoprotein A and apolipoprotein-B improved in both groups in a positive direction. However, cardiorespiratory capacity and the serum levels of high-sensitive C-reactive protein and Lipocalin-2 decreased, while irisin and paraoxonase 1 increased significantly, but only in Group 1. Conclusions: Six weeks of aerobic training, regardless of its intensity, could induce favorable changes in functional tests, body composition and serum lipids, even in severely obese, extremely unconditioned patients in both groups. However, moderate-intensity aerobic training should at least increase cardiorespiratory capacity and yield a better lipid profile oxidative status and inflammation profile.

## 1. Introduction

The World Obesity Federation declared obesity as a chronic, relapsing and progressive disease [1]. Obesity is considered to be a significant public health burden worldwide and represents a major determinant of global morbidity, mortality and disability risk of musculoskeletal disorders, hypertension, cardiovascular diseases (CVDs) and metabolic disorders, such as Type 2 diabetes mellitus and dyslipidemia [2]. According to a recently published WHO report, obesity has reached pandemic dimensions, as its prevalence nearly tripled between 1975 and 2016 [3]. Overweight affects 2.5 billion adults, of whom 890 million are clinically obese. Unfortunately, adolescent obesity has quadrupled since 1990 [4]. Moreover, morbid obesity (body mass index [BMI] over 40 kg/m^2^) has been associated with impairment of cardiorespiratory fitness and muscle strength, limiting the capacity to perform activities of daily living and increasing the economic costs associated with healthcare in this population [5,6]. The prevalence of morbid obesity is highest among people aged 40–59 years (11.5%), followed by people aged 20–39 (9.1%) and over 60 (5.8%) [7]. It is also estimated that severe obesity will be a bigger problem than underweight by 2025 [7].

Until the 1980s, white adipose tissue (WAT) was considered the sole energy store, while its diverse endocrine and immune functions were unknown [8]. It is now clear that besides serving as a major depot, WAT secretes a plethora of adipo(cyto)kines, making the bed for several comorbidities associated with obesity [9,10,11]. Independently from their metabolic effects, every known adipokine is involved in inflammation and immune response [10]. The imbalance in the secretion of pro- and anti-inflammatory adipokines causes low-grade chronic inflammation [11]. Acting through autocrine, paracrine and endocrine settings, adipokines affect cytokine and chemokine secretions as well as hormonal and growth inputs, and interact with lipid and carbohydrate metabolism [9]. The high-density lipoprotein (HDL)-associated antioxidant enzyme paraoxonase-1 (PON1) plays a role in several human diseases, including diabetes mellitus and atherosclerosis [9,12]. PON1 is able to inhibit the oxidation of low-density lipoprotein (LDL) and protect against its proinflammatory activation in the vessel wall by destroying biologically active lipids in ox-LDL. PON1 also reduces ox-LDL uptake by macrophages and stimulates HDL cholesterol outflow from these cells [13,14]. Decreased PON1 activity and adipokine imbalance may serve as predictive biomarkers for cardiovascular complications in obesity-related conditions [11,15,16].

Indeed, leptin has proinflammatory and adiponectin anti-inflammatory properties [17,18]. Serum leptin concentration is directly proportional to body fat and is considered as a satiety hormone [16]. Moreover, the adipose tissue of obese subjects secretes increased amounts of adipo(cyto)kines, like tumor necrosis factor α (TNF-α) and interleukin-6 (IL-6), which stimulates the hepatic production of proinflammatory acute-phase proteins, such as C-reactive protein (CRP) [19]. The concentration of elevated CRP is associated with the elevated risk of cerebrovascular diseases and several neoplasias [20]. Obesity is commonly linked with high ox-LDL levels, producing changes in the metabolism of visceral adipocytes, which are involved in the regulation of insulin resistance [21]. Also, there is a close correlation between BMI, waist circumference, fat percentage, triglyceride, homeostatic model assessment–insulin resistance (HOMA–IR) and lipocalin-2 (LCN2), as increased serum levels of LCN2 have been found in obese adolescents and in adult patients [22]. LCN2 is expressed in atheromatous plaques and affects the pathogenesis of plaque development and stability. LCN2 accelerates atherosclerosis with the enhancement of inflammatory responses, and plasma levels are nearly correlated with various metabolic and inflammatory parameters, including hsCRP [23,24].

Exercise training is an effective strategy to improve insulin sensitivity and thus reduce the risk of Type 2 diabetes mellitus and mitigate chronic inflammation [25]. Lifestyle modifications, including beneficial changes in dietary habits and physical exercise, have been shown to be effective for long-term weight loss and the prevention of weight gain. Moderate physical activity (150 and 200 min/week) may be needed to accelerate long-term weight loss and prevent weight regain [26]. Acute episodes of exercise increase the levels of proinflammatory adipokines, but chronic endurance physical activity reduces the circulating levels of these biomarkers [25]. The study of Wang et al. certified that serum concentrations of LCN2 are closely associated with obesity and its related chronic inflammation and metabolic complications [22]. Subsequent studies have demonstrated that high-intensity interval training, walking and running significantly decrease LCN2 concentrations at maximal heart rate (HR max) [26,27]. Contrary to the above-mentioned, the effects of one session of intense short-term incremental exercise have resulted in significantly increased LCN2 levels in obese and normal-weight men [28]. Also, in the study of Nakai et al., the LCN2 levels were stable over time and were not modulated by exercise or diet in healthy or overweight/obese people [29].

The direct measurement of VO_2_ max is the best method to measure cardiorespiratory fitness, indicating the amount of oxygen supplied to the respiratory, circulatory and muscular systems [30]. Cardiorespiratory fitness is often reduced in obese individuals; however, there is still no consensus on the best forms of exercise and their safety [30].

In the physical exercise and morbid obesity study by Fonseca Junior et al., the majority of exercise programs employed moderate-intensity physical activity [31]. It is important to note that, despite numerous data demonstrating the benefits of moderate-to-high-intensity exercise training in obesity, there are no data about the effect of low-intensity training on such individuals with extremely low capacity. Also, little information is available about the best sustainable and feasible form of physical exercise concerning the intensity to be advised for individuals with morbid obesity, as these patients are usually not able to perform high-to-moderate-intensity aerobic and strength training or exercise. Data are also controversial regarding the relationship of the form and intensity of the training with biochemical parameters, including adipokines.

In order to develop appropriate physiotherapy for morbidly obese, highly unconditioned patients, we need to evaluate the best feasible training intensity and discover how this exercise would affect obesity-related physiological and biochemical abnormalities. Therefore, we aimed to provide a multidimensional evaluation of the role of low- and moderate-intensity training on various anthropometric and cardiorespiratory parameters, as well as on the markers of atherosclerosis, oxidative stress and metabolic derangements, including serum levels of adipokines.

## 2. Materials and Methods

### 2.1. Study Design and Patients

A randomized controlled clinical study was conducted with the participation of patients referred to the Department of Physical Medicine and Rehabilitation, Faculty of Medicine, University of Debrecen, Debrecen, Hungary. All the patients signed a written informed consent form. The study was designed on the basis of CONSORT criteria (Consolidated Standards of Reporting Trials), as described elsewhere [32]. Ethical permission was provided by the Hungarian Medical Research Council 24949-1/2017/EKU.

During the prescreening period, patients underwent a complete history revision and physical examination. Patients also underwent an ambulatory examination and dietary counseling about 3 months prior to the study. Their diet was stable for at least 12 weeks preceding the study until its end. Daily protein supplementation was 1 g/body weight kg [33]. Telemedicine-based follow-up consultations were also provided to improve diet adherence (Figure 1).

Eligible subjects were obese patients who had consented to participate in the study, had a BMI > 35 kg/m^2^, were 18–70 years old and were able to undergo an ergospirometry test according to the WHO protocol without any cardiorespiratory signs or symptoms, and were able to walk with or without devices. There was no secondary cause of obesity (e.g., endocrine disease or medications). Further inclusion criteria were the following: bearing endurance, understanding the commands (based upon the Mini Mental State examination test’s complex task) and having no dementia, with the sum of the points exceeding 23. Resting blood pressure was ≤140/90 mmHg and the ejection fraction was over 40% during echocardiography. Patients withdrawing consent to participate, or having unstable cardiopulmonary conditions (i.e., myocardial infarction or cardiac arrhythmia within 3 months), untreated diabetes mellitus, alcohol dependency, other chronic neurological diseases such as Parkinson’s disease or multiple sclerosis, musculoskeletal diseases or presenting symptoms of peripheral arterial disease, untreated depression or chronic pain (visual analogue scale exceeding 5) were excluded.

The participants who consented and fit the inclusion criteria (number (n) = 40) were randomized into two groups, Group 2 (G2) and Group 1 (G1), by an independent person. We used a blocked randomization form. 20 patients were randomized to the G1 group (n = 20) and 20 patients to the G2 group (n = 20).

### 2.2. Interventions

Exercise tolerance was evaluated by a blinded assessor at the beginning and at the end of the training using ergospirometry (Piston Ltd., Budapest, Hungary, SN: 101-E0D-2014-011). All the tests were performed on a calibrated electromechanically broken bicycle ergometer (Ergometer EBike Basic and BP, ergoline GmbH, Bitz, Germany, SN: 2014004807). During ergospirometry, participants were asked to keep a cadence of 55 to 65 revolutions per minute. The test was terminated when patients reached the anaerobic threshold defined by a respiratory exchange ratio (RER) greater than 1.1 or volitional exhaustion. RER is the ratio of the carbon dioxide output to the oxygen uptake (VCO_2_/VO_2_). Termination criteria also included ST elevation ≥ 1.0 in leads without preexisting Q waves because of myocardial infarction (other than aVR, aVL or V1), a drop in systolic BP of ≥10 mmHg, central nervous system symptoms including ataxia, dizziness, syncope of worsened perfusion (cyanosis) and sustained ventricular tachycardia or other arrhythmias (atrioventricular blocks 2nd or 3rd degree).

The test was conducted in accordance with the ACSM guideline, and the target pulse rate was calculated at first assessment according to the Karvonen equation (HR target = /(HR max − HR rest) × 0.4 − 0.6)/+ HR rest. At G1, our goal was to achieve 40–60% of heart rate reserve (HRR). Target pulse rate at G2 was 30–39% of HRR. During ergospirometry, we also measured VO_2_ max level, which is a good marker of cardiorespiratory fitness. The WHO protocol was applied during the test as an incrementally increasing exercise by 25 W/min after a two-minute warm-up period. The test ended with a two-minute cooldown period and was symptom-limited. During the testing, the measurements included resting heart rate (HR rest) and maximum heart rate (HR max) by electrocardiogram (EKG), blood pressure (BP), oxygen uptake (VO_2_), carbon dioxide output (VCO_2_), VO_2_ max, VO_2_-VT, ventilation per minute (VE), load time, exercise time and respiratory ratio (RER). VO_2_ max was calculated as the highest 20 s average VO_2_ during the exercise test. VCO_2_, VO_2,_ ventilation per minute (VE) and RER were continuously monitored during exercise tests using breath-by-breath respiratory gas analysis. VO_2_-VT is identified when a change in the slope of the relationship between VO_2_ and VCO_2_ occurs. The ergospirometry automatically established the regression lines and their crossing points [34].

Both groups used pedometers to measure daily activity during the prescreened period and the six-week program. Body components (body fat, body fat% and lean body mass) of both groups were measured by a blinded assessor using In Body 720 on the first and last days of the therapy.

Peripheral blood samples from both groups were withdrawn after a 12 h overnight fast. After centrifugation, routine clinical parameters (i.e., creatinine, uric acid, glucose, hemoglobin A1c (HbA1c), triglyceride, total cholesterol, LDL-C and HDL-C) were measured with a Cobas 6000 autoanalyzer (Roche Ltd., Mannheim, Germany) in the Department of Laboratory Medicine, University of Debrecen, Faculty of Medicine, Debrecen, Hungary. High-sensitivity C-reactive protein (hsCRP), ApoA1 and ApoB levels were determined by immune-turbidimetric assays. All measurements were performed according to the manufacturers’ recommendations. Adipokine determinations were carried out with enzyme-linked immunosorbent assay (ELISA); samples were kept at −80 °C before the measurements. We determined serum levels of tumor necrosis factor (TNF-α), irisin, adiponectin, leptin, LCN2, chemerin and vaspin using Human TNF-α Quantikine HS Elisa (Cat: HSTA00E), Irisin ELISA (RAG018R), Human Leptin Quantikine ELISA Kit Cat: DLP00, Lipocalin-2/NGAL Human ELISA (RD191102200R), Human Chemerin Quantikine ELISA (Cat.: DCHM00) and Human Vaspin ELISA (RD191097200), respectively. All the patients underwent laboratory tests one day before and on the 42nd day of the therapy. A blinded assessor, who was an experienced physical trainer, measured body mass index (BMI), waist circumference, six-minute walking test (6MWT), timed up and go (TUG) test and apnea test one day before and on the 43rd day for both groups. Waist circumference was measured at the top of the hip bone and the bottom of the rib during normal exhalation [35].

The examination was carried out as an inpatient form under strictly controlled conditions in a group setting. Thus, the supervision, inspection and control of the patients took place exactly on time. Motivation of the patients and adherence to the diet were easier to achieve in an inpatient setting.

Both groups’ (G1, G2) inpatient training was performed on a 5-day per 6-week basis (30 consecutive weekdays). G1 patients participated in conventional, customized physiotherapy for 3 × 30 min, including a 30 min breath training, at a low intensity. This was followed by 30 min of moderate-intensity aerobic training and 30 min of resistance training. G2 patients participated in conventional, customized physiotherapy for 3 × 30 min, including a 30-min breath training, at a low intensity. This was followed by 30 min of moderate-intensity aerobic training and 30 min of resistance training. Each training form also contained 5 min warm-up exercises and 5 min cooldown exercises. From the second week, a fourth type of training (bicycle exercise) was added for the remaining five weeks. Patients performed aerobic training on bicycles (Christopeit, Top-Sports Gilles, Velbert, Germany, SN: DE18272186), aiming to reach personal target pulse rates. Target pulses were 40–60% of HRR at G1 and 30–39% of HRR at G2 during the exercise period, and training was symptom-limited with a warm-up (5 min), a therapy (20 min) and a cooldown (5 min) phase. For resistance training, patients in both groups used 50–60% of a one-repetition maximum (RM) weight with 8–12 repetitions. The average calorie intake was 1312. 20 ± 71.65 kcal daily for 12 weeks prior to the study and, to provide stable conditions for evaluation, did not change during the investigation period. The diet that patients had followed at home for the previous 3 months was similar.

### 2.3. Outcome Measurements

Evaluations were performed at baseline and 6 weeks later. HR was checked during cycling sessions every 5 min (Handheld Pulse Oximeter, Guangdong Biolight Meditech Co., Ltd., Zhuhai, China, SN: M01EO18751). Blood pressure was monitored before and after the session (Rextra, F.Bosch Practicus, Bisingen, Germany, SN: K140623).

To determine the level of physical activity intensity, we used the Borg Rating of Perceived Exertion (RPE) during therapy. Ranging from 6 (no exertion) to 20 (maximum exertion), RPE is based on the physical sensations that a person experiences during physical activity, including increased heart rate, increased respiration or breathing rate, increased sweating and muscle fatigue. (Borg, 1998) [36].

The load capacity of the patients was assessed by ergospirometry, and VO_2_ max, respiratory VO_2_-VT, load time, maximal exercise capacity (HR max) and the metabolic equivalent (MET) were determined. Load time is the time until the patient endures ergospirometry charging. Additionally, we measured functional independence measures (FIMs) and assessed the global functional capacity of the therapy [37].

In addition, anthropometric data were analyzed and body consumption analysis (body fat, lean body mass, body fat%), functional tests, 6MWT and TUG tests were performed. We also investigated serum lipid parameters (total cholesterol, triglyceride, HDL-C, LDL-C, Apo-A, Apo-B levels, oxLDL, hsCRP, concentrations and calculated HOMA–IR. HOMA–IR is the most commonly used insulin resistance index, a model of the relationship between glucose and insulin dynamics under fasting conditions to predict insulin resistance, simplified by the formula: fasting insulin (μU/dL) × fasting blood glucose (mmol/L)/22.5 [38].

Serum adipokine levels of leptin, TNF-α, chemerin, vaspin, adiponectin, LCN2 and irisin were studied and PON1 activity was measured.

### 2.4. Statistical Analysis

STATISTICA (ver: 8.0; StatSoft Inc., Tula, OK, USA) software was used for data analysis. The statistical tests were non-parametric with mean and median values, considering the small sample sizes. Data were presented using a descriptive analysis (mean ± standard deviation or medians [lower quartile–upper quartile]. Differences before and after the exercise program were determined using Wilcoxon matched-pairs tests for data with non-normal distributions. The Mann–Whitney U-test was used for intergroup analysis. Results were considered significant at the level of *p <* 0.05.

## 3. Results

After prescreening, 40 patients were randomized into two groups [G1 (n = 20), G2 (n = 20)] and all of them completed the program. There were no significant differences in the characteristic data of the two groups (Table 1). Pedometer data indicated that there was no significant change in the daily activity of the patients in either group during the prescreening and study periods. There was no significant difference in the nutrition of the two groups. The only important difference between G1 and G2 was that G1 received moderate-intensity aerobic training while G2 received low-intensity aerobic training. The low-intensity aerobic training was the same as traditional physiotherapy.

### 3.1. Change in Body Composition, Functional Tests and Aerobic Capacity

By the end of the training program, anthropometric parameters (BMI, waist circumference, body fat, body fat%) were significantly reduced (*p* < 0.001) in both groups. All functional tests (6MWT, TUG, apnea test) improved significantly (*p* < 0.001) in both groups. MET increased significantly (*p* < 0.001) in both groups. Aerobic capacity tests (VO2 max, load time and VO_2_-VT/kg) were increased significantly (*p* < 0.05) only in G1 (Table 2 and Table 3).

### 3.2. Alterations in CVD Risk Factors and Adipokine Concentrations

LDL-C and cholesterol concentrations decreased significantly (*p* < 0.05), while HDL-C levels in both groups showed a significant increase (*p* < 0.05) by the end of the program. Triglyceride and TNF-α levels remained unchanged. The levels of oxLDL (*p* < 0.05) decreased significantly, but only in G1. Apo–A and Apo–B levels decreased significantly (*p* < 0.05) in both groups. HOMA–IR also decreased significantly (*p* < 0.05 and *p* < 0.05 respectively) in both groups. Circulating concentrations of leptin (*p* < 0.05) significantly decreased in both groups. But it was only in G1 that circulating concentrations of LCN2 were significantly decreased (*p* < 0.05). The circulating concentration of irisin (*p* < 0.05) increased significantly, but only in G1. Levels of adiponectin, chemerin and vaspin showed no significant changes in either group. CRP concentrations were also found to be significantly reduced upon the completion of the program (*p* < 0.05), but only in G1. PON-1 (*p* < *0*.05) enzyme levels showed a significant increase, but also only in G1 (Table 4).

### 3.3. The Most Clinically Significant Findings

-Only moderate-intensity aerobic training improved VO_2_ max, which clearly indicated an improvement in cardiorespiratory fitness.-Both groups showed improvements in the 6MWT. The ability to walk is a well-established index of functional capacity and quality of life. The improvement in the 6MWT results in both groups may indicate an improvement in walking ability and, consequently, in quality of life.-Body composition was positively affected by both moderate- and low-intensity aerobic training.-The favorable development of serum lipids leads to favorable changes in arteriosclerosis.-Decreases in waist circumference occurred as a result of both low- and moderate-intensity aerobic training. Excess fat carried around the waist can increase the risk of high blood pressure, high blood cholesterol, heart disease and diabetes mellitus Type 2. Reducing the waist circumference is important for the improvement and prevention of heart disease and diabetes mellitus Type 2.-Leptin decreased with both low- and moderate-intensity training, but irisin and LCN2 decreased only with the moderate-intensity training. It may be worth following the intensity of the training with changes in irisin and LCN2.

## 4. Discussion

Six weeks of low-and moderate-intensity aerobic training, combined with resistance training, was effective in improving cardiorespiratory functions, body composition, cardiovascular metabolic risk factors and the levels of some adipokines in morbid obese patients. Despite the latest recommendations for physical activity in obesity management, where 30–60 min of moderate-to-high-intensity training is suggested for most days of the week [39], in cases of patients with extreme obesity and very low cardiorespiratory capacity we have to face a great challenge because severely unconditioned morbidly obese subjects with low VO_2_ max are unable to accomplish or maintain higher intensity training. The study of Rejeki et al. used moderate-intensity aerobic training forms for obese females. They found that moderate-intensity aerobic training, combined with resistance training, is effective in improving body composition and adipokine levels [40]. To the best of our knowledge, this is the first study to evaluate and compare cardiorespiratoric functions with ergospirometry during low- and moderate-intensity training using several anthropometric and cardiorespiratory parameters as markers of atherosclerosis, including inflammation, metabolic derangements, serum levels of lipoproteins and adipokines. We combined aerobic training with resistance training, as the latter form of exercise improves neuromuscular adaptation and prevents sarcopenia; it also increases muscle strength without increasing VO_2_ max [41,42].

Among the investigated adipokines, the level of serum leptin decreased the response of low- and moderate-intensity aerobic training too [43]. Decreased serum leptin is mediated by a decrease in adipose tissue, which is characterized by a decrease in BMI and fat mass. Leptin is a suitable biomarker for managing and evaluating exercise interventions for unconditioned severely obese persons. Our results confirmed the results of Maturana et al. [44]. We found significant alterations in the levels of serum LCN2 and irisin, which changed to a favorable direction in response to six weeks of moderate-intensity aerobic training. This suggests that LCN2 and irisin might be sensitive markers of the metabolic milieu for subjects with morbid obesity during such exercise training forms. This is in agreement with previous studies indicating a significant positive relationship between serum LCN2 levels and body mass, body fat, body fat percentage and hsCRP. Indeed, Wang et al. found that higher concentrations of LCN2 correlated positively with BMI, waist circumference and body fat percentage in obese subjects [22]. Weight loss reduced LCN2 levels and provided beneficial outcomes for markers of inflammation and oxidative stress. Indeed, the activity of the HDL-associated antioxidant PON1 enzyme increased significantly, indicating an improvement in protection against atherosclerosis and cardiovascular diseases due to low- and moderate-intensity aerobic training forms [11]. The adipokines and their altered secretions link obesity and its several consequences, including atherosclerosis and cardiovascular morbidity [11].

Both low- and moderate-intensity aerobic training improve body compositions, anthropometric parameters and functional tests, but a load of appropriate intensity is required to reach significant improvements in cardiorespiratory fitness.

As for the intensity of the training, previous studies utilized moderate-to-high-intensity aerobic training in obese patients. Our trial proved that low- and moderate-intensity aerobic training may also result in favorable changes, even in morbid obesity. The load intensity is important to achieve appropriate favorable changes in adipokine levels and load capacity. Six-week low-intensity training offers enough to diminish body weight, improve functional capacity and decrease the consequences of arteriosclerosis in morbidly obese patients. But six-week moderate-intensity aerobic training is needed to increase cardiorespiratory fitness with increased VO_2_ max and yield a better lipid profile, both of which contribute to longer-term beneficial effects. Aerobic fitness, which can commonly curb the performance of daily activities, is traditionally determined by VO_2_ max, requiring patients to be maximally exerted and motivated [45]. Improved aerobic fitness and increased VO_2_ max may reduce the consequences of obesity and help to maintain the favorable changes provided by the training programs.

## 5. Conclusions

These data demonstrate that both low-and moderate-intensity aerobic exercise programs, along with stable dietotherapy and resistance training, improve endurance and the markers of atherosclerosis, oxidative stress, inflammation and metabolic derangements in morbidly obese patients. Our study suggests that low-intensity physical activity in patients with extreme obesity may be potentially effective in promoting favorable changes in morbidly obese patients in whom achieving moderate intensity is required to achieve favorable changes in some adipokines and develop cardiorespiratory fitness. Consequently, their cardiovascular risks decrease and their quality of life improves. Previous studies examined moderate-to high-intensity aerobic training, but we have proved that similar changes can be achieved with low- and moderate-intensity training. Previously, the effects of low- and moderate-intensity aerobic exercise programs on body composition, cardiorespiratory function, biochemical risk factors and adipokine levels were not evaluated in a complex manner in morbidly obese patients.

## 6. Implication for Clinical Practice

It is important to emphasize the numerous health benefits to be gained with higher physical activity, even at a low intensity, in persons with extreme obesity. To achieve the effects of weight and fat loss, the implementation of exercise training programs in persons with extreme obesity should primarily aim to increase physical fitness, reduce cardiometabolic risk and improve quality of life.

It is worth continuing the research to see how the results achieved in 6 weeks can be maintained in the long term and how regular exercise can be incorporated into the lives and everyday routines of extremely obese patients who previously led a sedentary lifestyle.

## 7. Limitation

Due to our limited number of inpatient beds, we were able to investigate a relatively small number of patients. In previous studies, the duration was generally longer, but the weekly frequency was less and outpatient forms were used. We chose 6 weeks because our patients did not tolerate more than 6 weeks of inpatient form. Data measured in an inpatient setting allowed for accurate measurement, but limited comparability with data measured in an outpatient population. Multicentered studies involving more obese subjects may further corroborate our findings. In addition, a longer study period and extended lengths of training programs may provide even more favorable and permanent changes.

The World Medical Association (WMA) Declaration of Helsinki—Ethical Principles for Medical Research Involving Human Subjects—was fully adhered to during the research.

## Figures and Tables

**Figure 1 nutrients-16-04251-f001:**
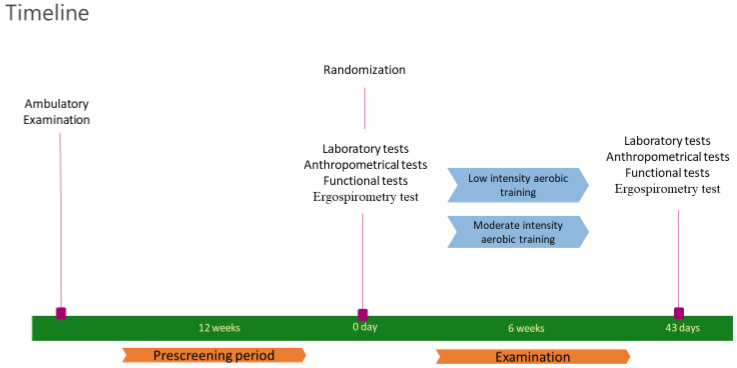
Timeline.

**Table 1 nutrients-16-04251-t001:** Patient characteristics.

	Group 2	Group 1
number (n)	20	20
age (year)	59 [45–68]	56.61 [40–68]
women: men ratio	16:4	13:7
diabetes mellitus Type 1/Type 2	2:5	3:3
sleeping apnea	4	2
lipid-lowering agents	6	5
walking independently: walking with medical aids (ratio)	13:7	15:5
sedentary lifestyle vs. regular physical activity (>3 × 30 min/day)	20:0	20:0
comorbid cardiovascular conditions (n)	18	15
comorbid mental health condition (n)	3	2
obesity in family history	12	8
calorie intake (kcal)	1312.21 ± 71.65	1312.21 ± 71.65

Abbreviations: n—number.

**Table 2 nutrients-16-04251-t002:** Change in body composition.

	Group 2 (n = 20)			Group 1 (n = 20)			
	Admission	Discharge	Inside Group Significance (*p*)	Admission	Discharge	Inside Group Significance (*p*)	Between Group Difference (*p*) *
Body weight (kg)	128 [84.5–186]	121 [83–170]	0.00	112.93 [72.1–151.0]	107.5 [70.1–143.5]	0.00	0.23
BMI (m^2^/kg)	43.91 [3.65–54.2]	42.09 [35.04–51.9]	0.00	41.57 [33.3–52.09]	39.5 [32.8–49.01]	0.00	0.23
body fat (%)	51.05 [37.00–56.80]	50.01 [35.50–56.30]	0.00	47.34 [30.7–58.0]	46.3 [25.7–58.3]	0.00	0.75
body fat (kg)	63.1 [36.3–99.2]	59.02 [34.1–90.00]	0.00	56.48 [28.9–87.7]	53.07 [23.8–84.9]	0.00	0.61
lean body mass (kg)	32.45 [18.3–48.3]	31.44 [18.4–45.5]	0.00	33.64 [20.7–53.4]	32.61 [19.9–46.2]	0.00	0.7
HOMA–IR	6.35 [0.45–30.43]	4.31 [0.18–23.12]	0.00	6.32 [0.79–29.06]	5.26 [1.21–41.5]	0.00	0.54

Abbreviations: n—number, kg—kilogram, m—meter, BMI—body mass index, HOMA–IR—homeostatic model assessment–insulin resistance. Significance level was *p* < 0.05, -median (IQR)- with Wilcoxon matched-pairs tests. Mann–Whitney U-test *.

**Table 3 nutrients-16-04251-t003:** Change in functional tests and aerobic capacity.

Average (Standard Deviation)	Group 2 (n = 20)			Group 1 (n = 20)			
	Admission	Discharge	Inside Group Significance (*p*)	Admission	Discharge	Inside Group Significance (*p*)	Between Group Difference (*p*) *
VO_2_ max average (mL/kg/min)	12.38 [6.86–16.56]	13.58 [6.69–17.48]	0.14	13.82 [6.48–18.38]	19.27 [12.43–25.34]	0.00	0.00
MET	2.31 [1.6–3.5]	3.36 [1.7–5.1]	0.00	4.34 [1.8–6.3]	5.97 [3.6–7.4]	0.00	0.00
Load time (min)	8.01 [2.57–14.29]	8.82 [1.47–15.19]	0.06	8.71 [2.55–13.16]	11.22 [7.17–15.32]	0.00	0.00
VO_2_-VT/kg (mL/min/kg)	7.6 [3.85–13.56]	8.56 [4.84–12.8]	0.12	8.78 [4.69–12.95]	9.45 [5.14–15.62]	0.00	0.78
6MWT (s)	322.8 [182–460]	381.52 [281–477]	0.00	367.19 [98–537]	432 [193–580]	0.00	0.74
TUG (s)	9.37 [6.53–20.7]	7.35 [5.32–11]	0.00	9.12 [5–26]	7.01 [5–13]	0.00	0.94
Apnea test (s)	31.54 [15–48.09]	44.7 [25–76]	0.00	40.32 [19.6–69]	51.75 [33–81]	0.00	0.57
Waist circumference (cm)	135.54 [103–160	130 [100–150]	0.00	123.14 [105–146]	119 [100–145]	0.00	0.13
FIM	118 [109–124]	121 [113–125]	0.00	122 [113–126]	124 [118–126]	0.00	0.67

Abbreviations: n—number, *p*—level of significance, VO_2_ max—maximal oxygen consumption, MET—metabolic equivalent, VO_2_-VT—ventilatory threshold, 6MWT—6 min walking test, TUG—timed Up and Go, FIM—Functional Independence Measure. Significance level was *p* < 0.05, -median (IQR)- with Wilcoxon matched-pairs tests. Mann–Whitney U-test *.

**Table 4 nutrients-16-04251-t004:** Change in CVD risk factors and adipokine concentrations.

Average (Standard Deviation)	Group 2 (n = 20)			Group 1 (n = 20)			
	Admission	Discharge	Inside Group Significance (*p*)	Admission	Discharge	Inside Group Significance (*p*)	Between Group Difference (*p*) *
cholesterol (mmol/L)	4.78 [2.8–7]	4.16 [2.6–6.8]	0.00	4.99 [2.8–7.5]	4.56 [2.4–6.8]	0.00	0.36
triglyceride (mmol/L)	1.8 [0.6–5]	1.6 [0.9–3.4]	0.95	2.26 [0.7–7.2]	2.03 [0.6–5.7]	0.64	0.23
HDL-C (mmol/L)	1.21 [0.8–1.7]	1.04 [0.7–1.4]	0.00	1.12 [0.8–1.6]	1.21 [0.8–3.7]	0.00	0.10
LDL-C (mol/L)	3.04 [1.2–5.1]	2.6 [1–4.7]	0.00	3.32 [1.1–5.2]	3.00 [1–4.7]	0.047	0.42
oxLDL (U/L)	63.78 [30.97–86.12]	60.7 [29.03–77.32]	0.52	77.4 [24.99–113.8]	70.91 [21.7–117.33]	0.03	0.001
Apo–A (g/L)	1.47 [1.21–1.93]	1.25 [1.03–1.64]	0.00	1.44 [1.01–2.5]	1.24 [0.92–1.57]	0.00	0.21
Apo-B (g/L)	1.02 [0.66–1.6]	0.93 [0.62–1.59]	0.01	1.44 [1.01–2.5]	1.32 [0.92–1.57]	0.014	0.75
hsCRP (mg/L)	9.48 [0.6–19.89]	8.61 [1.1–20.32]	0.32	8.56 [1–24]	6.14 [0.5–22.4]	0.05	0.001
TNF-α (pg/mL)	0.93 [0.31–5]	0.95 [0.31–5]	0.26	0.63 [0.3–1.19]	0.65 [0.15–1.26]	0.13	0.35
Lipocalin 2 (ng/mL)	87.57 [36–148.17]	82.3 [28.32–166.17]	0.54	77.60 [31.56–129.51]	59.63 [44–123.87]	0.01	0.04
Irisin (μg/mL)	2.69 [1.55–5.8]	2.73 [1.93–5.1]	0.8	2.84 [1.92–4.24]	3.43 [2.93–5.11]	0.014	0.01
Chemerin (ng/mL)	148.62 [94.36–233.26]	147.89 [95.36–208.25]	0.95	131.15 [58.77–188.16]	124.43 [49.95–184.07]	0.24	0.37
Vaspin (ng/mL)	0.37 [0.03–1.82]	0.35 [0.03–1.69]	0.48	0.33 [0.03–0.9]	0.32 [0.03–1.34]	0.39	0.1
Adiponektin (μg/mL)	5.81 [1.4–11]	6.75 [2.17–13.18]	0.06	6.15 [1.56–19.3]	6.1 [1.87–20.77]	0.7	0.11
Leptin (ng/mL)	83.65 [24.04–141.85]	64.76 [22.76–108.91]	0.00	78.48 [9.14–159.73]	57.97 [7.09–125.18]	0.00	0.85
PON-1	32.1 [8.42–70.93]	40.48 [10.31–140.09]	0.052	53.34 [16.32–132.02]	63.57 [18.25–193.92]	0.04	0.00

Abbreviations: n—number, *p*—level of significance, HDL—high density lipoprotein-cholesterol, LDL-C—low density lipoprotein-cholesterol, Apo—apolipoprotein A, Apo-B—apolipoprotein B, CRP—C reactive protein, TNF-α-tumor necrosis factor alpha, PON-1—Paraoxonase 1, Significance level was *p* < 0.05, -median (IQR)- with Wilcoxon matched-pairs tests. Mann–Whitney U-test *.

## Data Availability

The original contributions presented in the study are included in the article, further inquiries can be directed to the corresponding author.

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
