# Peer review of "Effect of Low- and Moderate-Intensity Aerobic Training on Body Composition Cardiorespiratory Functions, Biochemical Risk Factors and Adipokines in Morbid Obesity"

_nutrients, 2024, doi:10.3390/nu16234251_

Round 1

Reviewer 1 Report

Comments and Suggestions for Authors

The manuscript is well-written and understandable.

The research topic of physical activity is relevant for conceptual research, and also for clinical practice in obesity.

However, I am wondering whether the manuscript fits well into the Nutrients journal, given that the study did not consider nutrition or diets.

The sample size of 40 individuals is very small. Has an a priori power analysis been conducted? Are estimates reliable with such a small sample?

The specific randomization procedure could be described in more detail.

Were there differences between females and males? Given the hormonal differences, differences in muscle mass, etc. this may have affected the results.

Likewise, age differences can have an influence given the decrease of muscle mass and cardiovascular fitness with aging.

The conclusion that physical activity had an effect is not valid. It could simply be driven by random effects over time. There is no passive control group without physical activity to control for such time effects.

The conceptual contribution could be better highlighted. How do the results advance the literature? Already a large number of studies exists on physical activity interventions in overweight individuals.

The practical relevance could be highlighted with more detailed examples from everyday life.

Were there any health constraints for the individuals? Risks of performing demanding physical activity for the cardiovascular system for the most vulnerable participants?

Author Response

Dear Reviewer

Thank you very much for your time and work to deal with my manuscript.

At the Department of Physical Medicine and Rehabilitation, Faculty of Medicine, University of Debrecen, we have been treating obese and extremely obese patients since 2011. Since then, we have been measuring and analysing the basic data of our patients. In collaboration with the Division of Metabolic Diseases, Department of Internal Medicine, Faculty of Medicine, University of Debrecen, we have the opportunity to conduct more detailed analysis of metabolism and body composition in addition to measuring their cardiorespiratory function. The number of patients studied was influenced by the lenght of the study. Many patients did not agree to be hospitalised for 6 weeks or could not be included in the study at all due to associated diseases.

We used a block randomisation form. The sample size was under 100, so we couldn’t use simple randomisation. Thehad no prior knowledge of the block size. An independent person randomly assigned patients into 2 groups.

The age and gender of patients can also affect body composition. Our female patients were in their menopause. Both male and female patients had inactive lifestyles and were significantly overweight. The female gender was predominant in the study population. Due to the small sample size, we did not separate our results by gender, as this would have resulted in a small number that would not be statistically significant.

The literature recommends high- and moderate- intensity aerobic training for obese and extremely obese patients, but due to the extremely low exercise capacity, it was not feasible in the present study population. The aim of the study was to show that low intensity aerobic training, also have practical benefits, induce favourable changes in functional tests, body composition and serum lipids even in severely obese extremely unconditioned patients. However, moderate-intensity aerobic training is necessary to develop cardiorespiratory fitness.

Our patients had joint pain, some of them also used assistive devices. Both groups had diabetes as a comorbidity, but not cardiorespiratory disease. All of them were able to participate in the surveys and ergospirometry tests. However, their results undoubtedly show that their exercise capacity was extremely low.

Reviewer 2 Report

Comments and Suggestions for Authors

General Overview

The manuscript addresses the effects of low and moderate-intensity aerobic training on body composition, cardiorespiratory functions, biochemical risk factors, and adipokine levels in morbidly obese individuals. The research is relevant given the global burden of obesity and its associated comorbidities. However, there are areas where the manuscript can be improved in terms of clarity, methodology reporting, and discussion depth.

Detailed Feedback

Title

Current Title: "Complex evaluation of the effect of supervised low and moderate intensity aerobic training..."

Suggestion: Simplify the title for clarity. For example, "Effects of Low and Moderate-Intensity Aerobic Training on Body Composition and Adipokines in Morbid Obesity: A Randomized Controlled Trial."

Abstract

Strengths: Summarizes the study design and key results effectively.

Weaknesses:

Results are presented in overly technical terms without highlighting practical implications.

Lacks a sentence on the clinical significance of findings.

Suggestions:

Rephrase: “Functional abilities improved in both groups; however, cardiorespiratory capacity increased significantly only in Group 1” to emphasize the clinical importance.

Introduction

Strengths: Provides a comprehensive background on obesity and associated metabolic consequences.

Weaknesses:

Over-reliance on older references. Incorporate recent studies, especially post-2020.

Some redundancy in describing adipokines and their role in obesity-related inflammation.

Suggestions:

Streamline the discussion on adipokines by integrating references more concisely.

Add more recent global data or systematic reviews to support the claim of obesity being a pandemic issue.

Methods

Strengths: Provides detailed descriptions of patient recruitment and training protocols.

Weaknesses:

Missing information on sample size calculation, which is critical for validating statistical power.

The description of outcome measures (e.g., Borg Scale, VO2 max) is overly technical and needs simplification for a broader audience.

Suggestions:

Include a paragraph explaining the rationale for the sample size and statistical methods used.

Clarify if the study was registered (e.g., clinical trial registration).

Results

Strengths: Provides comprehensive data tables with statistical analysis.

Weaknesses:

The tables are dense and difficult to interpret without clear summaries in the text.

No mention of effect sizes, which are important for understanding the magnitude of changes.

Suggestions:

Add effect size data for key outcomes.

Summarize the most clinically significant findings in bullet points in the text.

Discussion

Strengths: Links findings to previous studies effectively.

Weaknesses:

The discussion on adipokines like LCN2 is overly detailed and may confuse readers unfamiliar with the topic.

The implications for clinical practice are not well addressed.

Suggestions:

Reorganize the discussion to prioritize findings with the greatest clinical relevance, such as improvements in cardiorespiratory fitness and inflammatory markers.

Add a subsection on "Clinical Implications" to discuss how these findings could influence exercise recommendations for morbid obesity.

Limitations

Strengths: Acknowledges key limitations such as sample size and study duration.

Weaknesses:

The potential bias from the inpatient setting and the exclusion of severely comorbid patients is not addressed.

Suggestions:

Discuss how the inpatient setting may have enhanced adherence and whether this could limit generalizability to outpatient populations.

References

Strengths: Broad range of studies cited.

Weaknesses:

Some references are outdated (e.g., from the 1990s).

Suggestions:

Replace older references with more recent meta-analyses or large-scale studies from the last five years.

Specific Recommendations for Improvement

Language and Readability:

Simplify technical jargon for non-specialist readers while maintaining scientific rigor.

Review sentence structures for clarity and conciseness.

Figures and Tables:

Use clearer labeling and add annotations or highlights to emphasize key findings.

Ethical Considerations:

Explicitly state patient confidentiality measures and data-sharing policies.

Future Directions:

Recommend exploring the long-term sustainability of these training regimens.

Author Response

Dear Reviewer,

Thank you very much for your time and work to deal with my manuscript.

At the Department of Physical Medicine and Rehabilitation, Faculty of Medicine, University of Debrecen, we have been treating obese and extremely obese patients since 2011. Since then, we have been measuring and analysing the basic data of our patients. We designed the research based on the data of these patients. In collaboration with the Division of Metabolic Diseases, Department of Internal Medicine, Faculty of Medicine, University of Debrecen, we have the opportunity to conduct more detailed analysis of metabolism and body composition in addition to measuring their cardiorespiratory function. The number of patients involved in the study was influenced by the length of the study. Many patients did not agree to be hospitalised for 6 weeks or could not be involved in the study at all due to comorbidities.

Thank you very much for your constructive advice. Thank you for your suggestion on the title, I have taken it and changed it. In a sentence, I mentioned that, unfortunately, obesity has quadrupled at a young age, which is worrying in relation to adulthood. I re-edited the introduction to make it easier to understand logically. I took it and incorporated what I could. I updated the relevant literature from the past years. The statistical tests were nonparametric with mean and median values, considering of the small sample sizes The tables contain a lot of data, I highlighted the significant results with colour for easier interpretation. I summarized the most clinically significant findings in bullet points. I emphasised the relevant clinical implications. In the future, it would be worthwhile to follow up the patients in a new study to confirm the long-term results of the treatment. We only have sporadic data on the patients. We have examined some of them in an outpatient setting, but these data are not suitable for reporting evidence-based results.
